# The Effect of Deep and Slow Breathing on Retention and Cognitive Function in the Elderly Population

**DOI:** 10.3390/healthcare11060896

**Published:** 2023-03-20

**Authors:** Su-Ha Lee, Dae-Sung Park, Chang-Ho Song

**Affiliations:** 1Department of Physical Therapy, Sahmyook University College of Health Science, 26-21, Gongneung2-dong, Nowon-gu, Seoul 01795, Republic of Korea; dmddo1009@naver.com; 2Department of Physical Therapy, Konyang University College of Health Science, 158, Gwanjeodong-ro, Seo-gu, Daejeon 35365, Republic of Korea; daeric@konyang.ac.kr

**Keywords:** cognitive function, cognitive skill, deep and slow breathing, geriatric, retention

## Abstract

The purpose of this study was to apply deep and slow breathing to the elderly, who can be classified as potential dementia patients, to confirm changes in the cognitive functions of learning and memory. Forty-five elderly subjects were randomly and evenly divided into a rest group (RG), a before group (BG), and an after group (AG). Measurements of their cognitive abilities were obtained before testing (PT), 30 min after learning (STT), and 24 h after learning (LTT). After PT measurements were obtained from all three groups, the RG and AG conducted new cognitive skills learning, while the BG performed deep and slow breathing (DSB) for 30 min before learning new cognitive skills. After all the three groups underwent 30 min of learning, the STT was performed. Subsequently, the AG performed DSB for 30 min. Finally, 24 h after learning, the LTT was conducted for all three groups. Changes were compared and analyzed by measuring the retention of new cognitive skills and attention, working memory, and spatial perception of cognitive functions. A two-way repeated measure analysis of variance measured the effect of the application of DSB in the three groups. These results demonstrated a significant interaction of time and time*group in all measurements of retention and attention, working memory, and spatial perception. This study confirms the benefit of DSB as part of a dementia prevention training protocol.

## 1. Introduction

With aging, deteriorations in health functions, including physical, cognitive, and cardiopulmonary functions, and social activity occur. Dementia is attracting attention as a disease to be watched most in old age. Dementia caused by aging occurs in 80–90% of degenerative brain diseases; moreover, aging is associated with a general decline in various functions of the brain, such as memory impairment, speech impairment, loss of spatio-temporal ability, and changes in personality and emotions [1].

Approximately 15% of all cases of dementia can be fully treated through early intervention, which highlights the importance of early diagnosis and proper treatment to slow down the progression of the disease. Currently, along with technological advances, various diagnostic tests and treatment programs are being actively implemented in the field of dementia diagnosis and management. While there have been advances in diagnostic and treatment programs for dementia, including technological developments, there is a need for training programs not only for patients with dementia but also for older adults who are at risk of developing the disease [2].

Recently, research activities from various angles have been reported so that the elderly can correctly perform all health function promotion activities, including cognitive function, and information exchange and discussion are being conducted to maintain the hu-mane quality of life with maximum independence in the given environment by the elderly themselves. It is actively being carried out [3].

However, there is a lack of development of individual cognitive function training tools or training equipment, and it is difficult for individuals to use the developed equipment autonomously at home, resulting in a lack of clinical use. In addition, most of the cases are used with the help of a professional trainer by visiting an institution rather than using it at home. In addition, it was reported that the dropout rate was high because the developed cognitive function training program was complexly designed or the subject’s understanding and concentration on the program were low due to problems such as difficulty setting [4].

Therefore, many researchers are paying attention to the development of cognitive training programs that can be easily followed and maintained by elderly persons. Among them, the most popular training technique is breathing control. Reportedly, the ability to detect space and three-dimensional movements is greatly improved with the application of DSB [5]. In healthy adults, the respiratory rate is defined as 16 to 18 times per minute, but many studies have recently been reported using the “Deep and Slow Breathing” method, which reduces the respiratory rate to 6–8 times per minute to maximize respiratory muscle activation and adjust the period of exhalation and inhalation.

The analysis of neural signals throughout the brain during deep and slow breathing showed the same active regions of the brain during new perceptual-motor skills. The activation areas were prefrontal cortex, post-parietal cortex, limbic system, cerebral motor cortex, and central sulcus [6].

The prefrontal cortex performs thought processing and inhibition, such as perception, short-term memory, executive memory, and regularity, the peripheral system is responsible for procedural learning, skill memory, and emotion, and the posterior parietal cortex controls planned movement, spatial perception, and attention.

This suggests that there is a close interaction between deep and slow breathing, cognitive function performance, and task learning process [7].

In addition, several studies have demonstrated that deep and slow breathing affects vagus nerve excitation and improves the retention of skill learning. The excitation of the vagus nerve stimulates the release of neurochemicals such as acetylcholine, epinephrine, and brain-derived neurotrophic factor (BDNF), which have been reported to have positive effects on neurogenesis, neuroplasticity, and neuronal repair [8,9].

This means that a simple breathing method can quickly promote the recovery of skills lost due to brain disease or neurological problems and can be expected to be clinically useful for improving cognitive function performance and tasks or skills learning [7].

The learning of new skills or the relearning of skills lost due to neurological problems go through the process of acquiring permanent skills through sequential steps.

At this time, in order to judge whether the learning was successful, it can be judged by re-measuring the skills acquired through previous practice or experience and evaluating the ability to re-perform the skills as accurately as possible. This ability is called retention [10].

Previous studies have confirmed the retention ability between deep and slow breathing and perceptual-motor skill learning. Additionally, the active area of the brain discovered through the application of deep and slow breathing has been shown to control various cognitive functions [11,12]. However, there has been no study that accurately measured retention ability and verified short-term and long-term effects on cognitive skill learning. In addition, there has been no study to confirm whether it actually helps improve cognitive function.

Furthermore, there are no studies that have confirmed the effect of deep and slow breathing on retention ability and cognitive function in elderly individuals with dementia.

In addition, since previous studies included the application of deep and slow breathing as well as other exercise techniques during intervention, it is difficult to identify whether breathing only contributes to cognitive function improvement and the degree of breathing control.

Therefore, there is a need for a study to confirm the quantitative retention ability measurement and cognitive function of skill learning according to the application of deep and slow breathing, targeting community-dwelling elderly who are prone to exposure to senile dementia rather than healthy adults. To investigate the effects of deep and slow breathing, it is crucial to examine the newly acquired cognitive skills and functions.

In this study, we applied DSB to an elderly population, who can be classified as potential dementia patients, to confirm changes in learning and memory retention and cognitive function by area.

## 2. Materials and Methods

### 2.1. Participants

This study consisted of 45 subjects. The study’s inclusion criteria require participants to be 65 years or older, without any hearing or vision impairments, and with a score of 24 or higher on the Korean Simple Mental State Test (MMSE-K). The exclusion criteria involve individuals who have previously taken the color-word test (Stroop test), patients with cardiopulmonary disorders or difficulty with deep and slow breathing, or those with anemia or neurological disorders. Prior to beginning the experiment, the Konyang University Institutional Review Board approved this study with the IRB approval number KYU-2020-054-01.

### 2.2. Procedures

A priori sample size estimation was performed for a two-way repeated measures ANOVA using the G*Power software (version 3.1.9.4; Kiel University, Kiel, Germany) to estimate the sample size needed for a two-way repeated measures ANOVA. We input the following parameters: statistical test  =  ANOVA: repeated measures, between factors; effect size f  =  0.56; α err prob  =  0.05; power (1 − β err prob)  =  0.80; number of groups  =  3; number of measurements  =  3; Corr among rep measures = 0.5. This resulted in a total sample size of 24, which was appropriate for our study condition (n = 45). We determined the effect size by using the “retention test of cognitive skill test” from previous studies [7].

The community-dwelling elderly recruited 45 subjects for this study between 1 June and 15 June 2020. The selection and exclusion criteria were explained to the subjects, and they were randomized into three groups: rest group (RG), before group (BG), and after group (AG), with 15 subjects in each group. The measurements and interventions were conducted in the laboratory of the College of Health Science at Konyang University in Daejeon, South Korea. The retention of new cognitive skills and cognitive function by area was measured using Superlab, with three trials of each test: pre-test (PT), STT (30 min after learning), and LTT (24 h after learning). All three groups performed the PT, and then the RG and AG learned new cognitive skills using Superlab. In the BG, new cognitive skills were learned after applying deep slow breathing (DSB) for 30 min. After 30 min of learning, STT was performed in all three groups. Subsequently, the AG performed DSB for 30 min after STT. After 24 h, LTT was performed in all three groups (Figure 1).

#### 2.2.1. Intervention with Cognitive Skill Tasks

In this experiment, the Stroop test was redesigned to check the stimulus of the match condition and the disagreement condition randomly presented using colors and words; subjects were to press a button corresponding to the color of the letter [13]. The stimulus was presented for 2000 ms, and the feedback after the stimulus was provided as “correct answer” and “incorrect answer” for 800 ms. For task-learning (TL), a total of 10 sets (100 times) were performed. One set (10 times) was performed 30 min after the TL for the STT to check the ability to learn and retain memory for the task. One set (10 times) was also performed 24 h after the TL for the LTT. Retention was calculated by determining the correct answer rate for each set and the reaction time for each number of times.

#### 2.2.2. Intervention with Deep and Slow Breathing

For the subject to concentrate on breathing, an independent, isolated space was prepared where only the subject and the tester could enter, and a comfortable chair was arranged to create an experimental environment. A monitor was placed in front of the subject to provide a DSB video guide, and pre-training was performed with the video three times before training.

For DSB, an alternative non-breathing method was selected and performed 6–8 times per minute for 30 min using a 2:2:4 breathing cycle. The right nostril was covered and the subjects inhaled using the left nostril for 2 s. After holding their breath for 2 s, the subjects closed their left nostril and exhaled using their right nostril for 4 s.

#### 2.2.3. Measurement with New Cognitive Skills Retention

A computer-programmed cognitive technology task designed using Superlab (Ver 5.0.5, Cerdrus Co., San pedro, CA, USA) was used to measure the retention. The cognitive technology task was produced as a Stroop test [13]. At the initiation of the test, 48 stimuli with matching colors and words and 48 stimuli with disparate colors and words are set, for a total of 96 stimuli, which appear continuously and randomly.

At this time, the goal was to press a button with the same color, not meaning, of the letter. The time from the start of the stimulation until the subject pressed the button was automatically stored and recorded as the reaction time. The percentage correct was calculated as the ratio of the number of consonant stimuli correctly selected by the subject among the total target consonant stimuli. Each stimulus was shown for 2000 ms and the interval time between stimuli was randomly arranged at 800 ms.

#### 2.2.4. Measurement with Cognitive Function by Areas

For all tests of cognitive function, the time from the start of stimulation until the subject pressed the button was automatically stored and recorded as the response time. The percentage of correct answers was calculated as the proportion of the number of consonant stimuli correctly selected by the subject out of the total target consonant stimulus.

##### N-Back Test

To measure working memory, the “2-back” was devised and used as follows [14]: As testing commenced, one of the Hangul consonants appeared in the center of the screen for 250 ms. Then, the programmed consonants appeared continuously and randomly; if the same consonant appeared again with a single consonant stimulus in between, the subject’s task was to recognize it as a target and press the red button as quickly as possible. The entire task consisted of 200 consonant stimuli (30 targets and 170 non-targets); each stimulus was shown for 250 ms, and the interval time between stimuli was randomly arranged at 1500 ms.

##### Go/No-Go Test

To measure attention, a Go/No-go test was designed and used [15]. At the start of the measurement, one of the Hangul consonants appeared in the center of the screen for 500 ms. Then, the programmed consonants appeared continuously and randomly. The subject’s task was to recognize targets among the consonants and press the red button as quickly as possible. The total task consisted of 200 consonant stimuli (30 targets and 170 traps); each stimulus was shown for 500 ms, and the interval time between stimuli was randomly arranged at 20,000 ms.

##### Mental Rotation Test

To measure spatial perception, a mental rotation test was devised and used [16]. When the measurement is started, one out of five non-verbal characters appeared in the center of the screen for 2000 ms, followed by the random appearance of a series of programmed non-verbal characters. The subject’s task was to determine whether the reference stimulus was the same as or different from the displayed stimulus by comparing the reference stimulus displayed on the left side of the screen with the comparison stimulus, which varied according to the rotation angle displayed on the right side of the screen. The total task consisted of 40 stimuli (20 targets and 20 traps); each stimulus was randomly arranged and shown for 2000 ms.

### 2.3. Statistical Analysis

All data were analyzed using the IBM SPSS Statistics ver. 18.0 software (IBM CO., Armonk, NY, USA). We conducted a normality test using Shapiro–Wilk to determine if the dependent variable’s normal distribution was met with respect to the independent variable. After confirming that the normal distribution was met, we performed Mauchly’s sphericity test. If the sphericity assumption was met, we confirmed the significance probability of the within-subject effect test using univariate analysis. If the sphericity assumption was less than 0.05, we concluded that the assumption was not met, and we confirmed the significance probability using Wilks’ Lambda. A comparison within and between groups according to the time and application of DSB was performed using two-way repeated measure analysis of variance. When there were significant differences by time or by time group, post hoc analysis was performed for multiple comparisons between groups using Bonferroni’s test. For multiple comparisons of the three groups, 5% of the significance level was divided by 3 as a type 1 error, 1.7%, and α = 0.017 was set as the significance level. If the significance probability value of the time and group of each test was less than 0.017, it was interpreted that there was an effect of the factor, also if the significance probability value of the time*group test was less than 0.017, it was interpreted that there was an interaction.

## 3. Results

### 3.1. Characteristics of Subjects

The general characteristics of the subjects are shown in Table 1.

### 3.2. Change of New Cognitive Skills Retention

By confirming the change in the percentage correct and the reaction time in retention, significant differences were significant differences between groups by time or by time*group (*p* < 0.05).

The post hoc analysis for intra-group multiple comparisons revealed that in terms of the percentage correct, there was no significant difference in the RG between the tests. However, in the BG, there was a significant increase in the percentage correct in the STT and LTT compared with that in the TL (*p* < 0.017). In the AG, there was a significant increase in the percentage correct in the LTT compared with that in the TL and STT (*p* < 0.017). Regarding the response time, the RG showed a significant increase in the STT and LTT compared with that in the TL (*p* < 0.05). However, in the BG, response time was significantly decreased in the STT and LTT compared with that in the TL (*p* < 0.017), and in the AG, the response time was significantly decreased in the LTT compared with that in the TL and STT (*p* < 0.017) (Table 2).

### 3.3. Change of Cognitive Function by Area

As a result of confirming the change in the percentage correct and reaction time in retention, significant differences were significant differences between groups by time or by time*group (*p* < 0.05).

The post hoc analysis for intra-group multiple comparisons revealed that in terms of the percentage correct, there was no significant difference in the RG between the tests. However, in the BG, there was a significant increase in the percentage correct in the STT and LTT compared with that in the PT (*p* < 0.017). In the AG, there was a significant increase in the percentage correct in the LTT compared with that in the PT and STT (*p* < 0.017). Regarding the response time, the RG showed no significant difference between tests. However, in the BG, response time was significantly decreased in the STT and LTT compared with the LTT compared with that in the PT and STT (*p* < 0.017) (Table 3, Table 4 and Table 5).

## 4. Discussion

This study confirmed how well cognitive skills were maintained through controlled breathing performed before and after learning new cognitive skills, as well as the influence of DSB on cognitive function by area. As a result of the measurement, it was confirmed that there was an interaction between time and group and the influence of time and group factors on newly learned cognitive skills and cognitive functions. By evaluating the learning retention after 30 min of DSB in the before group (BG), this study revealed significantly higher retention in the BG compared to the groups that did not perform DSB before the learning activity (*p* < 0.017). Moreover, the AG, who performed DSB after the learning technique, also showed significantly higher retention after 24 h (*p* < 0.017). These results can be provided as basic data that can support the scientific mechanism of neurophysiological parameters according to the application of deep and slow breathing found in several previous studies. In previous studies, deep and slow breathing stimulated the vagus nerve to stimulate the production of neurochemicals, such as acetylcholine, norepinephrine, and brain-derived new management factor (BDNF). In relation to plasticity and nerve recovery, an effect for the recovery of neuroparalytic patients can be expected, and it plays a role as a potential medium for improving the retention of cognitive and motor skills learning [8,9,17,18].

In addition, we found active areas of the brain according to the application of deep and slow breathing [19,20,21,22,23,24]. Previous research has examined brain activity during deep and slow breathing using methods such as electroencephalography (EEG), near-infrared spectroscopy, oxygenated hemoglobin levels, functional magnetic resonance imaging (fMRI), and oxygen levels in the bloodstream (BOLD). Activation was found in the prefrontal cortex (Brodmann area 9, 10), posterior parietal cortex, and circumferential system area of the cerebral cortex, which are responsible for various cognitive functions. However, the long-term effects of deep and slow breathing on cognitive functions have not been extensively studied.

In this study, we aimed to investigate the effects of deep and slow breathing on newly learned cognitive skills and cognitive functions. Participants completed a color-word test and three cognitive function tests: the ‘N-back test’, ‘Go/No go test’, and ‘Mental Rotation test’. We measured the changes in cognitive skills and functions 30 min and 24 h after learning.

Our results show that deep and slow breathing significantly improved newly learned cognitive skills and cognitive functions (*p* < 0.017). Moreover, both the short-term and long-term effects of deep and slow breathing were confirmed in experimental group A (*p* < 0.017), and the long-term effect was confirmed in experimental group B (*p* < 0.017).

These findings suggest that simple breathing control techniques can be effective in improving the retention of newly learned cognitive skills and can inform studies investigating factors that promote cognitive skill learning and retention. Furthermore, our study provides a basis for future clinical studies exploring the effects of deep and slow breathing on cognitive function in older adults.

There are a few limitations to take note of in this study. Firstly, the sample size is relatively small, consisting of only 45 elderly participants, which may hinder the applicability of the results to larger populations. Moreover, the study design only permits a short-term evaluation of the impact of DSB on cognitive functions, with measurements taken at 30 min and 24 h after learning. The long-term impact of DSB on cognitive functions in potential dementia patients remains unknown. Furthermore, the cognitive task used to evaluate the effect of deep and slow breathing on newly acquired cognitive abilities is relatively basic and easy to learn. As a result, it is unclear whether similar outcomes would be seen with more complex cognitive tasks. Additionally, this study only examined the effects of one particular breathing method, and the potential advantages of other breathing control methods have yet to be examined.

Therefore, we would like to suggest the direction of future research that compares the effects by applying various technical task methods and breathing methods that convert the duration, intensity, and cycle of deep and slow breathing.

## 5. Conclusions

This study revealed that applying DSB can enhance the ability of elderly individuals to process new cognitive tasks and improve cognitive function. These findings suggest that deep and slow breathing training could serve as a simple yet effective method for developing cognitive training programs to prevent and manage dementia in older adults within the community. Furthermore, the ease with which this technique can be integrated into existing treatment protocols suggests it may have potential for use in cognitive rehabilitation programs. Ultimately, this study highlights the potential for slow breathing and cognitive training programs as a means of preventing and managing dementia in older adults.

## Figures and Tables

**Figure 1 healthcare-11-00896-f001:**
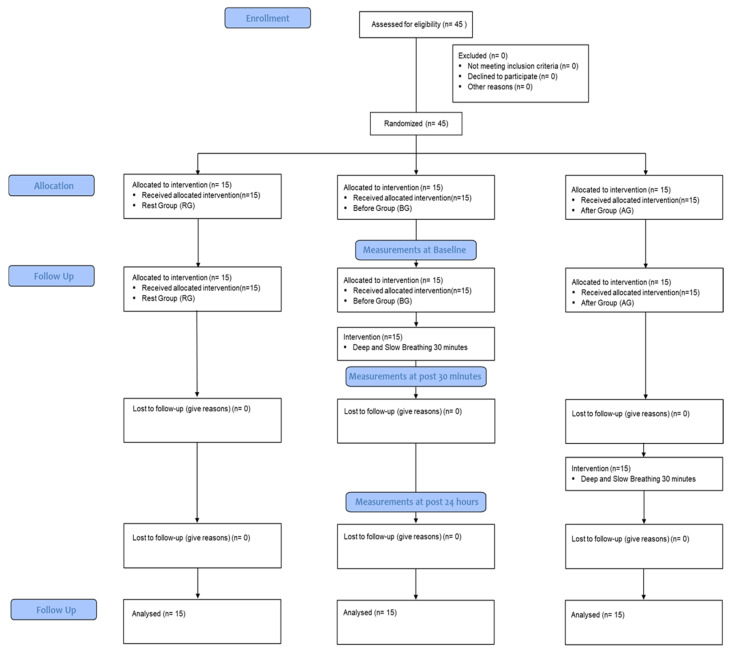
Flow chart.

**Table 1 healthcare-11-00896-t001:** Subject characteristics.

Characteristics	CG (n_1_ = 15)	BG (n_2_ = 15)	AG (n_3_ = 15)
Gender (male/female)	7/8	9/6	8/7
Age (years)	71.10 ± 4.87 ^1^	69.70 ± 5.10	71.30 ± 6.37
Height (cm)	168.54 ± 9.87	167.33 ± 6.75	168.93 ± 8.67
Weight (kg)	63.65 ± 9.65	67.39 ± 8.71	65.45 ± 7.65
BMI	22.55 ± 2.65	24.16 ± 2.49	23.19 ± 2.81
MMES-K (score)	25.43 ± 6.40	26.01 ± 5.01	25.09 ± 5.65

^1^ Mean ± standard deviation, CG: control group, BG: before group, AG: after group, BMI: body mass index, MMES-K: mini-mental state examination-Korean.

**Table 2 healthcare-11-00896-t002:** Change of new cognitive skill retention (N = 45).

Retention	TL	STT	LTT	*Post hoc*	*F*		
Correct (%)							
CG	50 ± 12.69 ^1^	59 ± 14.30	58.75 ± 19.27		Time	67.60 *	
BG	45 ± 4.71	81.5 ± 9.52	86.25 ± 7.29	TL < STT, LTT			
					Time *Group	14.53 *	
AG	51 ± 26.85	59 ± 20.11	89 ± 11.01	TL, STT < LTT			
Reaction Time (ms)							
CG	1052.13 ± 157.32	1160.95 ± 170.53	1186.16 ± 152.58	TL < STT, LTT	Time	48.44 *	
BG	1088.16 ± 88.52	894.47 ± 84.89	916.49 ± 91.24	STT, LTT < TL			
					Time *Group	87.16 *	
AG	1093.84 ± 154.59	1094.22 ± 151.16	803.45 ± 157.58	LTT < TL, STT			

^1^ Mean ± standard deviation, CG: control group, BG: before group, AG: after group, TL: task-learning, STT: short-term test, LTT: long-term test, *: *p* < 0.01.

**Table 3 healthcare-11-00896-t003:** Change of working memory of cognitive function by area (N = 45).

N-Back	TL	STT	LTT	*Post hoc*	*F*		
Correct (%)							
CG	59.14 ± 17.97 ^1^	40.27 ± 10.05	48.39 ± 14.31		Time	80.00 *	
BG	54.07 ± 24.99	70.78 ± 13.61	48.32 ± 13.87	TL < STT, LTT			
					Time *Group	34.70 *	
AG	60.73 ± 18.89	72.16 ± 12.79	82.37 ± 12.03	TL, STT < LTT			
Reaction Time (ms)							
CG	790.31 ± 88.83	1005.78 ± 116.46	1101.01 ± 121.39	TL < STT, LTT	Time	116.05 *	
BG	784.19 ± 89.54	689.25 ± 65.32	1070.65 ± 75.69	STT, LTT < TL			
					Time *Group	83.41 *	
AG	815.47 ± 87.28	682.24 ± 69.35	634.57 ± 52.30	LTT < TL, STT			

^1^ Mean ± standard deviation, CG: control group, BG: before group, AG: after group, TL: task learning, STT: short-term test, LTT: long-term test, *: *p* < 0.017.

**Table 4 healthcare-11-00896-t004:** Change of attention of cognitive function by area (N = 45).

Go/No-Go	TL	STT	LTT	*Post hoc*	*F*		
Correct (%)							
CG	59.14 ± 17.97 ^1^	40.27 ± 10.05	48.39 ± 14.31		Time	47.07 *	
BG	54.07 ± 24.99	70.78 ± 13.61	48.32 ± 13.87	TL < STT, LTT			
					Time *Group	34.27 *	
AG	60.73 ± 18.89	72.16 ± 12.79	82.37 ± 12.03	TL, STT < LTT			
Reaction Time (ms)							
CG	564.84 ± 120.56	774.93 ± 68.13	987.18 ± 138.84	TL < STT, LTT	Time	90.49 *	
BG	563.61 ± 122.20	567.30 ± 43.32	1003.64 ± 137.37	STT, LTT < TL			
					Time *Group	62.13 *	
AG	557.01 ± 144.88	570.87 ± 51.00	534.26 ± 53.00	LTT < TL, STT			

^1^ Mean ± standard deviation, CG: control group, BG: before group, AG: after group, TL: task learning, STT: short-term test, LTT: long-term test, *: *p* < 0.017.

**Table 5 healthcare-11-00896-t005:** Change of spatial perception of cognitive function by area (N = 45.).

Mental Rotation	TL	STT	LTT	*Post hoc*	*F*		
Correct (%)							
CG	59.14 ± 17.97 ^1^	40.27 ± 10.05	48.39 ± 14.31		Time	96.87 *	
BG	54.07 ± 24.99	70.78 ± 13.61	48.32 ± 13.87	TL < STT, LTT			
					Time *Group	70.75 *	
AG	60.73 ± 18.89	72.16 ± 12.79	82.37 ± 12.03	TL, STT < LTT			
Reaction Time (ms)							
CG	1298.84 ± 58.23	1409.59 ± 110.83	1455.36 ± 134.75	TL < STT, LTT	Time	76.59 *	
BG	1329.98 ± 85.21	1151.42 ± 62.59	1463.40 ± 132.51	STT, LTT < TL			
					Time *Group	63.12 *	
AG	1306.20 ± 69.06	1175.39 ± 79.84	964.01 ± 48.00	LTT < TL, STT			

^1^ Mean ± standard deviation, CG: control group, BG: before group, AG: after group, TL: task learning, STT: short-term test, LTT: long-term test, *: *p* < 0.017.

## Data Availability

The authors confirm that the data supporting the findings of this study are available within the article.

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
