# Peer review of "The Effect of Deep and Slow Breathing on Retention and Cognitive Function in the Elderly Population"

_healthcare, 2023, doi:10.3390/healthcare11060896_

Round 1

Reviewer 1 Report

62: How were the participants recruited?

62: Is there a sample size calculation? If not, the stuy should be marked as exploratory in the title and through out the text

82: maybe a figure with the experimental design would it make easier for the reader to understand what was done in each group

158: Line "Heights" is not readable in the provided PDF

155: There should be a heading for the results section

175ff: please provide effect sizes for all main results

- no drop-outs? 

- were all participants able to follow the protocoll?

- any side effects?

Author Response

Thank you for your astute comments and suggestions, which have been instrumental in shaping this article into a more effective and impactful piece of scholarship.

Reviewer 2 Report

The study aimed to compare three training protocols in older adults with dementia, and to verify the effects of deep and slow breathing on the cognitive functions of

learning and memory.

I consider the initiative interesting, follow my considerations, perhaps they can improve the quality of the manuscript:

Minor

1. I suggest using the term "older adult" instead of "elderly" throughout the text.

Major

1. The Introduction section still lacks a good 'justification' for carrying out this study. Please work on this in depth!

2. Methodology

2.1 Detail inclusion and exclusion criteria

2.2 Detail the year, place where the study was carried out, as well as the team involved in the procedures

2.3 Present normality test

2.4 Was a sample calculation performed? If not, justify the fact and/or present it as a limitation of the study

2.5 The Intervention protocol "could" be better presented so that other researchers can replicate the study (this can guarantee a good number of citations for your study and also for the journal Healthcare). Suggestion: create a Table showing the phases and actions, or include images of the procedures.

2.6 Present the procedures used to recruit the evaluated

2.7 Present all ethical procedures, and especially the study approval protocol number

Discussion:

1. Work this section a little further, really clarifying how training changed the underlying mechanisms of all cognitive functions.

1.1 In this sense, better specify/differentiate the relationship between underlying mechanisms of training with cognitive skills and attention, working memory, and spatial perception of cognitive functions (this is the goal of the study).

1.2*** Every investigation has several Limitations. This study is no exception to the rule. Therefore, include a set of points as limitations of the study, which can be added with suggestions for future studies.

1.3 Finally, the conclusion is still very succinct. May be slightly enlarged!

Author Response

(The authors gave the same response as above.)

Reviewer 3 Report

Concerning the article "The effect of deep and slow breathing on the retention and cognitive function when learning new cognitive skills in Elderly" I think that the research conducted was comprehensive and the results were well presented. The findings showed that deep and slow breathing can have a positive impact on the retention and cognitive function in elderly individuals when learning new skills. The article highlights the importance of incorporating breathing exercises into daily routines for optimal brain function. Overall, I highly recommend this article for publication.

Anyway, I would like the Authors to do a few revisions:

- Why did you treat all study group participants treated as potential dementia patients? Was the age abobe 65 the only criterion?

- In the Methodology: "All the subjects had experience with the color-word test (Stroop test) prior to the experiment" - what do the Authors exactly mean, did they have an assessment with this test prior to the study?

- Were the cardiopulmonary diseases the only exclusion criterion? Did you also consider anemia or neurological disorders?

Author Response

Thank you for your comments and suggestions that helped make this article more effective and impactful.

Reviewer 4 Report

I am very glad to have been asked to revise this manuscript. Presented work is potentially interesting. The authors are presenting a feasibility (I suppose) study to evaluate the effects of deep and slow breathing to the older adults. Deep and slow breathing (DSB) techniques, as a component of various relaxation techniques, have been reported as complementary approaches in various treatments. Breathing plays an important role in pain signalling and autonomic nervous system (ANS) activation, emotion regulation, acid/base balance, and anti-inflammatory processes.

Major concerns:

Unfortunately, the manuscript is not well written and there are many limitations to this research that may place restriction on the soundness of all the study.

The most severe concerns both the evidenced base literature and the reporting of methodology according to which authors have carried out their research. I had difficulty reviewing the manuscript because it does not specify several details. As a result, after several times reading, I regret to say that I am not certain exactly what was done and, specially, why.

This creates a circumstance where I cannot recommend publication of the paper in the current form. My recommendation to the Editor is that the current submission will not be accepted for publication. My sense is that the study may be of potential interest to the IJERPH readership.

In my opinion, the authors did not provide the level of scientific detail that would be expected elsewhere. This issue is compounded by the fact that authors do not report literature base, purpose, hypotheses, and research design. The physiological (and psychological) effects of deep and slow breathing techniques and of the underlying mechanisms in healthy humans need to be reported together with the physiological implications to the respiratory, cardiovascular, cardiorespiratory and autonomic nervous systems, with particular focus sympathovagal balance. The theoretical rationale of the experimental paradigm needs a deep clarification. There are several empirical mindful breathing programs, this needs to be mentioned and for the authors to situate their work with work that is already done because the question of "why this program" did not feel sufficiently addressed. What the probable effect underlying the intervention among cognitive skills? I suppose the effect of long-term potentiation, but authors failed to discuss any argument.

Second, no solid mention was made for recruitment and enrollment procedure, research design description was absent. What was the reason to consider the participants as potential dementia patients??? 

Third, I was surprised to see that authors in the discussion reported that “In addition, we found active areas of the brain according to the application of deep and slow breathing”, this should mean that the experimental paradigm has been associated with neuroimaging recording but this is not true.

Fourth, the conclusion “The positive effect of DSB on learning, memory retention, and cognitive function by area were demonstrated herein. These results support the possibility of slow breathing and a cognitive training program for the prevention and management of dementia in the elderly” cannot be proposed considering that their study is not longitudinal or controlled.

Equally important is that the authors attend to the IJERPH style and format requirements (I note that references section also report 17 doi or sites as references with the consequence that the number of references appear 37 whereas really it was 20.

I hope the authors will find my suggestions helpful to express your perspective more clearly in a further study.

Author Response

(The authors gave the same response as above.)

Round 2

Reviewer 2 Report

Dear all, after analyzing the healthcare-2184659 manuscript, I consider that the changes were made. Therefore, I am in favor of its publication.

Reviewer 4 Report

Thank you for the update and responses. I think the current manuscript is much readable. 

·         Authors failed to describe literature, evidence base on the topic and elements necessary for understanding the rationale of their study. They failed to address my first previous issue on literature base. On the contrary, the current manuscript is  logically more organized and reported more information and details that are able to help in the research question. Thus, I suggest to accept the manuscript.